# Sjögren syndrome/scleroderma autoantigen 1 is a direct Tankyrase binding partner in cancer cells

Harmonie Perdreau-Dahl[1,2], Cinzia Progida 3, Stefan J. Barfeld[4], Hanne Guldsten[1], Bernd Thiede[5], Magnus Arntzen 6, Oddmund Bakke 3, Ian G. Mills[4,7,8], Stefan Krauss[9] & J. Preben Morth 1,2,10 ✉

Sjögren syndrome/scleroderma autoantigen 1 (SSSCA1) was first described as an auto-antigen over-expressed in Sjögren's syndrome and in scleroderma patients. SSSCA1 has been linked to mitosis and centromere association and as a potential marker candidate in diverse solid cancers. Here we characterize SSSCA1 for the first time, to our knowledge, at the molecular, structural and subcellular level. We have determined the crystal structure of a zinc finger fold, a zinc ribbon domain type 2 (ZNRD2), at 2.3 Å resolution. We show that the C-terminal domain serves a dual function as it both behaves as the interaction site to Tankyrase 1 (TNKS1) and as a nuclear export signal. We identify TNKS1 as a direct binding partner of SSSCA1, map the binding site to TNKS1 ankyrin repeat cluster 2 (ARC2) and thus define a new binding sequence. We experimentally verify and map a new nuclear export signal sequence in SSSCA1.

[1] Membrane Transport Group, Centre for Molecular Medicine Norway (NCMM), Nordic EMBL Partnership, University of Oslo, P.O. Box 1137 Blindern, 0318 Oslo, Norway. [2] Institute for Experimental Medical Research (IEMR), Oslo University Hospital, Ullevål PB 4956 Nydalen, NO-0424 Oslo, Norway. [3] Centre for Immune Regulation, Department of Molecular Biosciences, University of Oslo, Blindernveien 31, 0371 Oslo, Norway. [4] Prostate Cancer Group, Centre for Molecular Medicine Norway (NCMM), Nordic EMBL Partnership, University of Oslo, P.O. Box 1137 Blindern, 0318 Oslo, Norway. [5] Department of Biosciences, University of Oslo, P.O. Box 1137 Blindern, 0316 Oslo, Norway. [6] Faculty of Chemistry, Biotechnology and Food Science, Norwegian University of Life Sciences, P.O. Box 5003, N-1432 Ås, Norway. [7] Patrick G Johnston Centre for Cancer Research, Queens University Belfast, Belfast, UK. [8] Nuffield Department of Surgical Sciences, Faculty of Medical Science, University of Oxford, John Radcliffe Hospital, Oxford, UK. [9] Department of Immunology and Transfusion Medicine, Oslo University Hospital, and Hybrid Technology Hub—Centre of Excellence, Institute of Basic Medical Sciences, Faculty of Medicine, University of Oslo, Oslo, Norway. [10] Enzyme and Protein Chemistry, Section for Protein Chemistry and Enzyme Technology, Department of Biotechnology and Biomedicine, Technical University of Denmark, Søltofts Plads, 2800 Kgs. Lyngby, Denmark. ✉email: premo@dtu.dk

Sjögren syndrome/scleroderma autoantigen 1 (SSSCA1), was initially described as autoantigen p27[1,2], and discovered in the late 1990s as a novel autoantigen overexpressed in Sjögrens syndrome and/or scleroderma patients[1]. The human *SSSCA1* gene is located on chromosome 11 (11q13.1) and encodes a small soluble protein of 21.5 kDa with a predicted N-terminal zinc ribbon domain type 2 (ZNRD2), and an unknown domain in the C terminus. The protein is predicted to be widely expressed in most normal tissues and to present an overall moderate expression level (Human Protein Atlas[3] and PaxDb[4]). Functionally, SSSCA1 is largely uncharacterized although it has been linked to mitosis and centromere association[1].

While the function of SSSCA1 remains unknown, information about SSSCA1 has emerged in multiple studies related to the Wnt signaling pathway, diverse solid cancers, and ubiquitination. SSSCA1 has been reported for its possible implication in the Wnt signaling pathway, as identified in mass spectrometric and proteomic studies as a potential binding partner of Tankyrases 1 and 2[5], as a target of the Tankyrase drugs XAV939[6] and G007-LK[7], and also as a target of the E3-ubiquitin ligase RNF146, which regulates Tankyrase protein levels and acts as a positive regulator of the Wnt signaling[8]. The Wnt signaling pathway is a crucial pathway in animals implicated in a variety of cellular processes including proliferation, differentiation, motility, survival, and apoptosis. Aberrant activation of this pathway often leads to cancer or other diseases, notably colorectal cancer for which more than 90% of the cases present an activating mutation[9]. In colorectal adenocarcinomas, SSSCA1 shows increased mRNA expression levels and has been identified as a key genetic marker for stroma activation and for upregulated pathway activity in colorectal adenoma-to-carcinoma progression[10,11]. Recently, SSSCA1 has been highlighted in numerous studies as a potential target gene and putative biomarker for breast cancer[12–14]. It has been associated with genomic instability at 11q and poor survival in both metastatic oral squamous cell carcinoma[15] and pediatric metastatic neuroblastoma[16]. SSSCA1 is also indicated as a potential risk variant for type 2 diabetes[17].

Finally, SSSCA1 has been linked to the ubiquitination pathway as a potential binding partner of the E3-ubiquitin ligases RNF146[8], RNF166[18] and HERC2[19], and of RAPGEF2, a guanine nucleotide exchange factor substrate of the E3-ubiquitin ligase component FBXW11[20]. A number of large scale studies also proposed that SSSCA1 is found ubiquitinated on four sites (K22, K64, K67, and K82)[21–24].

Even though this multitude of publications has identified SSSCA1 in their experiments, the information is mainly obtained from large scale indirect studies. In this work, we specifically aimed at characterizing SSSCA1 for the first time, to the best of our knowledge, at the molecular, structural, and cellular level. We describe the general domain organization of SSSCA1 and identify three distinct domains including an N-terminal zinc finger domain, a disordered region and a C-terminal domain and describe that this domain organization is found in most eukaryotes and is highly conserved. We have determined the crystal structure of what we believe is a novel and unique Cys4 zinc finger domain in the N-terminal end at 2.3 Å resolution, and with structural and bioinformatic information we were able to identify putative orthologs in most animals including invertebrates and fungi. The C-terminal domain has a dual function as we here show the inclusion of an unusual nuclear export signal (NES) that overlaps with a docking site to the poly-ADP-ribosyltransferase Tankyrase 1 (TNKS1). Indeed, we identify and verify TNKS1 as a direct binding partner of SSSCA1 in three cancer cell lines. We map the binding site of SSSCA1 on TNKS1 to the ankyrin repeat cluster (ARC) 2.

## Results

### The domain architecture of SSSCA1 is evolutionary conserved.
The identified homologous of SSSCA1 all encode a small soluble protein between 15 and 25 kDa composed of three distinct domains. An uncharacterized zinc-binding domain at the N terminus shows the highest degree of conservation and is annotated by the Pfam database[25] as the unique Auto_anti-p27 domain belonging to the ZNRD2 family. A less conserved region with notable variations in length between species links to the C-terminal region, this domain spanning segment is predicted to be an intrinsically disordered region (IDR) high in proline content. With a disorder prediction probability above 0.5 for at least 30 consecutive residues, SSSCA1 is classified as an intrinsically disordered protein. The C-terminal domain includes a coiled-coil motif and is expected to form two consecutive α-helices from secondary structure prediction (Fig. 1a).

Several sequence databases (NCBI BLAST, Ensembl, SwissProt) were searched to identify possible SSSCA1 orthologs, and revealed potential orthologs throughout the eukaryote kingdom including vertebrates, invertebrates, protists, and fungi. Orthologs of SSSCA1 are surprisingly missing in birds (class Aves) and insects (class Insecta) (Fig. 1b, c). To further investigate SSSCA1 conservation between species, sequence alignments and disorder prediction plots were performed and showed that the distinct domain order found for SSSCA1 homologs is highly conserved from human to yeast (Fig. 1b).

### SSSCA1 is a binding partner of Tankyrase 1 in cancer cells.
SSSCA1 has been shown to be overexpressed in several cancers, in particular colorectal cancer. One of the major causes of colorectal cancer is an aberrant regulation of the Wnt signaling pathway and the consequent activation of TCF/LEF target genes such as the pan-oncogene c-MYC[26]. As this signaling pathway and c-MYC are also altered in a variety of other cancers such breast, cervical, lung, and prostate[27] cancer, we investigated the common function of SSSCA1 in three different human cancer cell lines (HeLa cervical cancer cells, SW480 colorectal cancer cells, and LNCaP prostate cancer cells).

We utilized a technique of tandem-affinity purification coupled to mass spectrometry (TAP–MS, Fig. 2a) which considerably decreases background levels and thereby increases the confidence in the interacting protein complexes[28]. The human gene of *SSSCA1* was cloned in frame with an 8x His-GFP tag and was transiently transfected into the three cancer cell lines. Following 18 h of expression, the cells were lysed and subjected to the TAP–MS method to determine potential binding partners of SSSCA1 that would be recognized in all three cancer lines. Enrichment values were calculated for every protein co-purified with the His-GFP-SSSCA1 purification and the control purification to estimate the most abundant hits. The Venn diagram in Fig. 2b represents the enriched proteins in the different cell lines, with 93 proteins overlapping in at least two cell lines (Supplementary Data 1). DAVID[29], Gene Ontology[30], and STRING[31] databases were searched to identify the functions of these proteins. This analysis revealed clusters of proteins related to the broad biological processes of protein binding and chaperones, microtubules and centrosomes, and RNA binding and processing. However, only five similar proteins were identified in all three cancer cell lines and these include SSSCA1 as expected, Tankyrase 1 (TNKS1), γ-tubulin (TUBG1), and the poorly characterized rRNA methyltransferase MRM2 and membrane associated VAPA (Fig. 2c). From these potential partners, TNKS1 was the most promising target as it gave the most consistent results, as it was co-purified with SSSCA1 and identified in all MS experiments, and not identified in the control

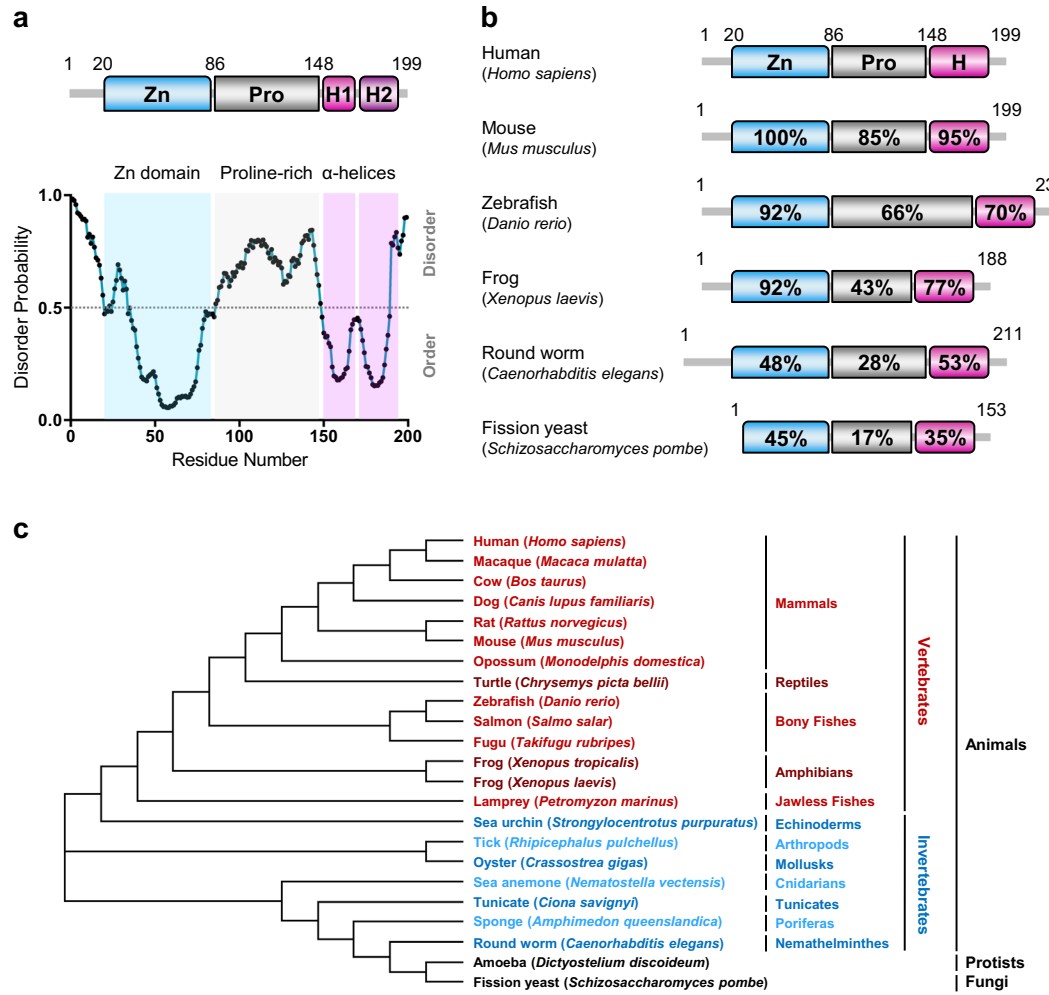

**Fig. 1 Schematic representation of the general domain organization of SSSCA1. a** Domain organization of human SSSCA1, aligned with a disorder probability plot and localization of the three domains. Numbers denote amino acid positions. Zn, N-terminal zinc-binding domain (residues 20–86). Pro, proline-rich region (residues 86–148). The C-terminal helical domain can be subdivided into two predicted α-helices H1 (150–167) and H2 (173–195). **b** Schematic domain organization of representative SSSCA1 orthologs. Zn, N-terminal zinc-binding domain. Pro, proline-rich region. H, C-terminal helical domain. Percentage of positive identities to human SSSCA1 is indicated. **c** Phylogenetic tree of SSSCA1. Orthologs' sequences retrieved from Ensembl, Genbank, or UniProt were aligned and a phylogenetic tree was constructed using the neighbor-joining tree method. Kingdoms, major phyla, and taxonomical groups are indicated. For clarity, vertebrate and invertebrate species are shown in red and blue nuances, respectively.

purifications. We further confirmed this interaction by immuno-blotting in all three cancer cell lines (Fig. 2d, full blots in Supplementary Fig. 5a). We also showed that both SSSCA1 and TNKS1 co-localize in HeLa cells during mitosis (Fig. 2e). These studies concluded that SSSCA1 is a binding partner of TNKS1, and the evidence for further in vitro characterization of the protein complex.

**Mapping of SSSCA1 and Tankyrase 1 potential binding regions**. The primary structure of SSSCA1 did not reveal any R-x-x-G-D-G consensus motif usually reported as the typical recognition motif of the TNKS1 ankyrin repeat domain (ARD) binding partners, even though deviations in the recognition motif can be tolerated[32]. In particular RNF146 has been shown to bind TNKS1 ARD without a clear consensus motif[33]. To locate the specific binding region, we generated a number of deletion constructs of GFP-tagged human SSSCA1 to identify the regions responsible for binding to TNKS1 (Fig. 3a). TAP from HeLa cells coupled to immunoblotting show that all SSSCA1 mutants lacking the C-terminal end helix H2 (which include constructs SSSCA1-Δ(Pro-

H2), -Δ(midPro-H2), -Δ(H1–H2), -ΔH2) were not able to maintain binding to TNKS1 during the purification (Fig. 3b, full blots in Supplementary Fig. 5b). While the Zn-binding domain and proline-rich region deletion mutants (constructs SSSCA1-Δ(Zn-Pro), -Δ(Zn)) pull downed TNKS1, indicating that these two domains do not play a role in the interaction. The modification of the structure and functionality of H2 by mutating the leucine residues to alanine (SSSCA1-H2$_{Leu/Ala}$) also abolished the binding to TNKS1, confirming the direct interaction of SSSCA1 and TNKS1 (Fig. 3b, full blots in Supplementary Fig. 5b) with the H2 helix region. To localize the binding region of SSSCA1 on TNKS1 we performed the opposite experiment and expressed several constructs of GFP-tagged human TNKS1 in HeLa cells (Fig. 3c). Here we observed that the TNKS1 mutant lacking the sterile alpha motif (SAM) and poly-ADP-ribose polymerase (PARP) domains still pulled down SSSCA1 (construct TNKS-Δ(SAM-PARP)), while the deletion of the ARD abolished the binding (construct TNKS-Δ(ARC1-ARC5)) (Fig. 3d, full blots in Supplementary Fig. 5c–d). More precisely, the deletion of TNKS1-specific subdomains, or ARCs, numbers 3, 4, or 5 did not affect the interaction with SSSCA1 (constructs TNKS-Δ(ARC3-PARP),

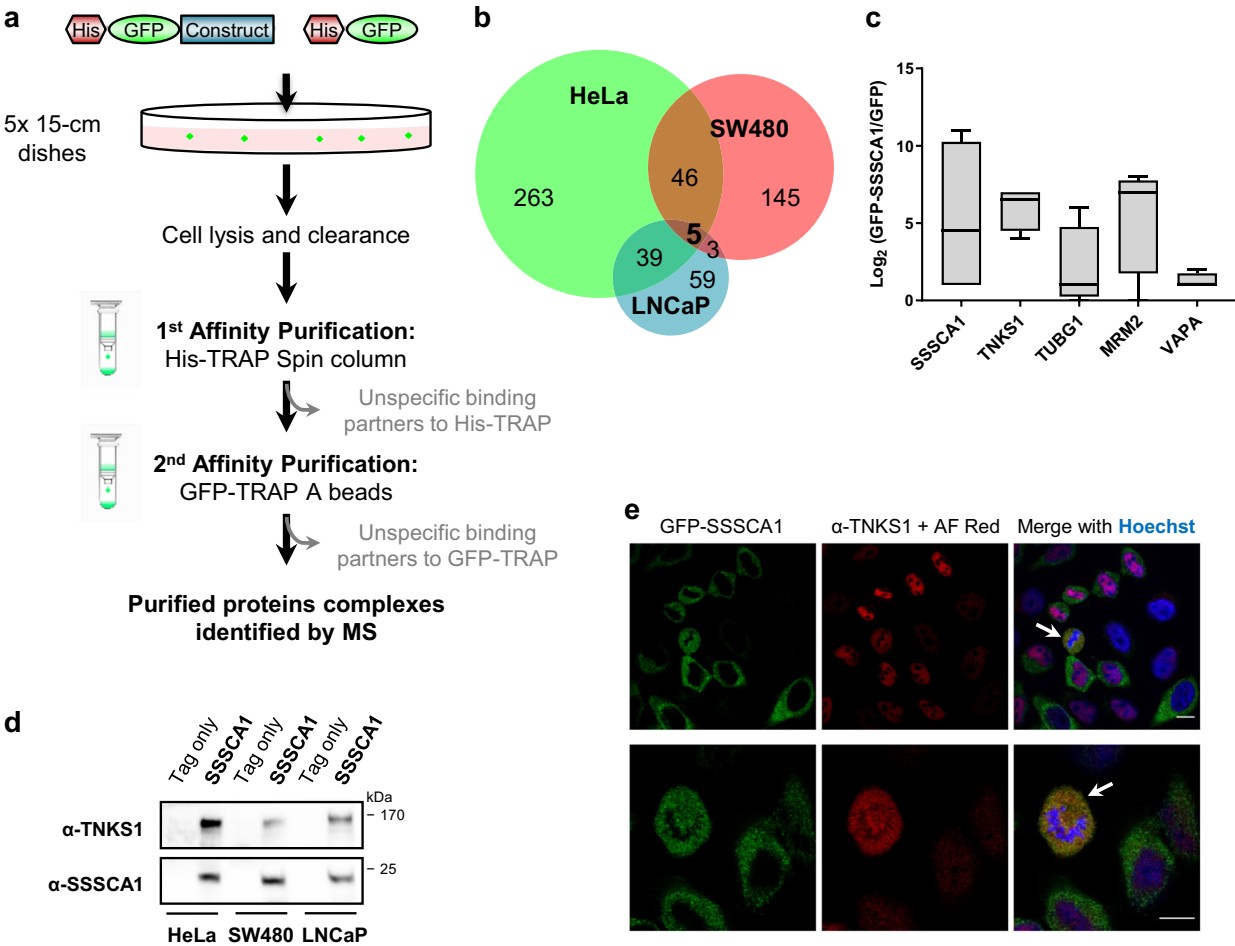

**Fig. 2 SSSCA1 is a strong binding partner of TNKS1 in cancer cells. a** Flow chart of the tandem-affinity purification coupled to mass spectrometry (TAP–MS) technique. **b** Venn diagram representing the significantly enriched proteins when purifying His-GFP-SSSCA1 in three different cancer cell lines. **c** Identification of five potential targets of SSSCA1 in the three cancer cell lines. **d** Confirmation of the identification of TNKS1 as SSSCA1's binding partner by immunoblotting. **e** Co-localization of His-GFP-SSSCA1 and endogenous TNKS1 during mitosis in HeLa cells. Two representative examples are shown. Arrows indicate co-localization. AF AlexaFluor. Scale bar: 10 μm.

TNKS-Δ(ARC4-PARP)). However, the mutants lacking ARC2 were not able to bind SSSCA1 (TNKS-Δ(ARC1-ARC4), TNKS-Δ (ARC2-PARP)) (Fig. 3d, full blots in Supplementary Fig. 5c–d), suggesting that this specific ARC is implicated in the direct binding to SSSCA1. Lastly, we purified human TNKS1's ARC subdomains 1–3 (ARC1–3) from *Escherichia coli* (Fig. 3e) and assessed its binding to FAM-tagged peptides of human SSSCA1 by fluorescence polarization (FP) (Fig. 3f). As shown previously[34], it was not possible to express and purify the single ARC2 domain without the flanking ARC1 and ARC3 domains. Despite variations in solubility between the H1 and H2 peptides, we could confirm that the H2 of SSSCA1 did interact with ARC1–3 while the H1 peptide showed lower affinity for ARC1–3, compared with the H2 peptide. The binding studies indicate some interaction with H1 with a measured affinity close to the affinity obtained for Axin1 positive control. The data accumulated for the H1 peptide were hampered by the low solubility and anisotropic polarization difference (Supplementary Data 2).

**SSSCA1 belongs to a novel family of Zn-binding proteins.** Human SSSCA1 N-terminal Auto_anti-p27 domain (residues 1–111, predicted MW 12.5 kDa) was successfully overexpressed in *E. coli* and purified over a Ni$^{2+}$-affinity and size-exclusion chromatography (SEC). The purified SSSCA1 construct was

predicted to form a dimer based on the elution time when compared with a set of common standard proteins after SEC (estimated native MW of 24.6 kDa, Fig. 4a). The Auto_anti-p27 domain was highly soluble and needed to be concentrated to more than 50 mg/ml to achieve crystallization success. The crystal structure of N-terminal SSSCA1 was determined at 2.3 Å resolution and phased using the single-wavelength anomalous dispersion method, based on the anomalous signals of two zinc ions (Fig. 4b, Table 1). The asymmetric unit contains two protomers that form an antiparallel dimer with a buried surface area of 1960 Å$^2$ ('Protein interfaces, surfaces, and assemblies' service PISA[35]), the interface is highly positively charged and is mainly formed by the helical part of the molecule. Each protomer binds one zinc ion coordinated by four conserved cysteines (human residues Cys53, Cys56, Cys70, and Cys73) structured around four antiparallel β-sheets (Fig. 4c). We were not able to model residues 1–12 and 78–111 in the electron density map and coincidentally, these regions correlate with the intrinsic disorder prediction plot of SSSCA1 and confirm that these residues likely are disordered in the protein crystal as well (Fig. 1a).

A systematic search for other proteins of similar architecture using available online services, the DALI server[36], PDBefold[37] revealed that SSSCA1 likely belongs to a novel Pfam family of Zn-binding ribbon domains (Auto_anti-p27, PF06677) which overall fold is not yet available in the protein data bank (PDB). However

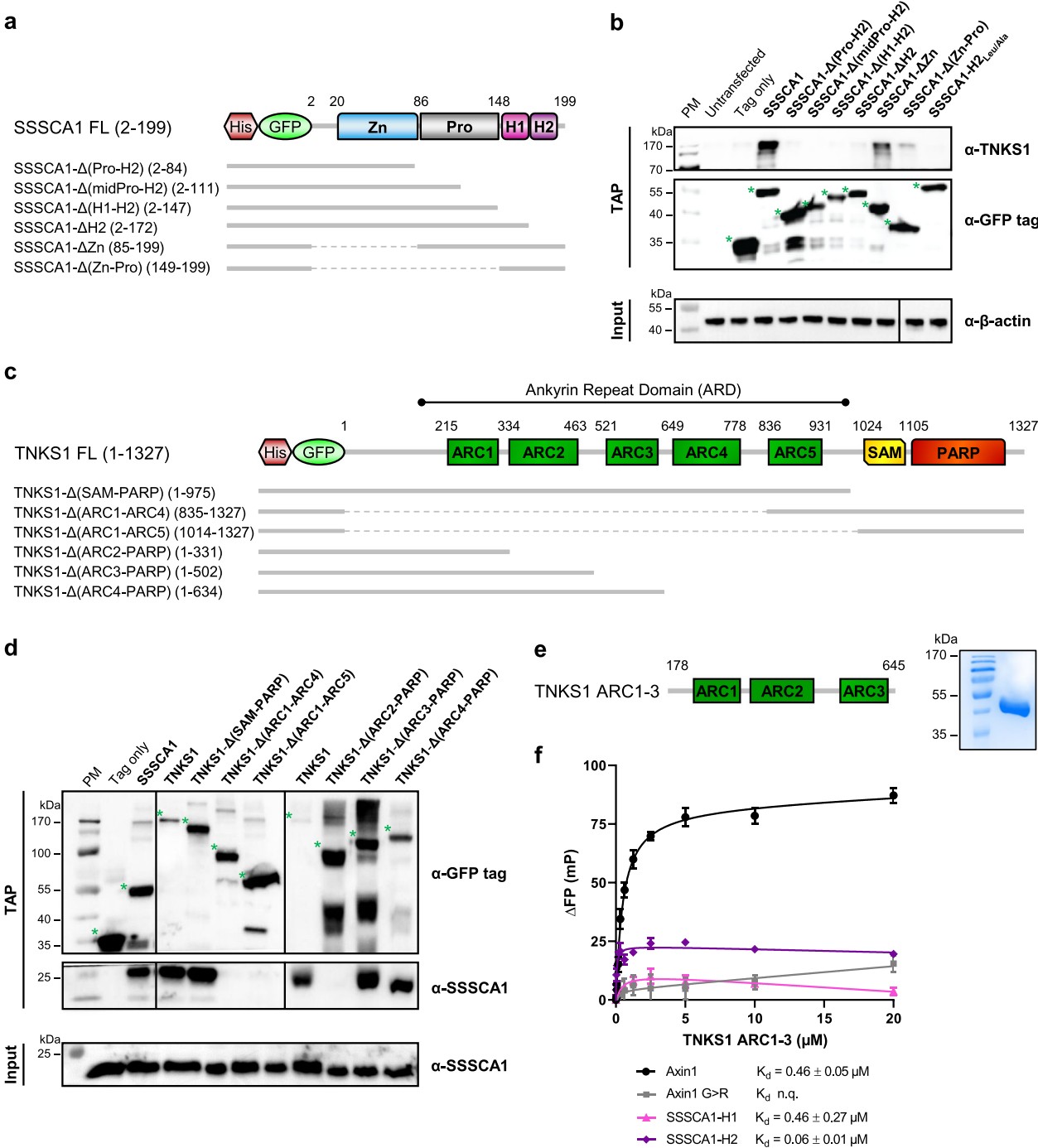

**Fig. 3 Mapping of SSSCA1 and TNKS1 binding. a** Schematic representation of His-GFP-tagged SSSCA1 constructs. FL full length. **b** Different constructs of human SSSCA1 were expressed in HeLa cells. After TAP, the binding of each construct to endogenous TNKS1 was assessed by western blotting. Green stars indicate the constructs of interest. β-actin was chosen for input control because of the very low and barely detectable endogenous expression of TNKS1. **c** Schematic representation of His-GFP-tagged TNKS1 constructs. FL full length. **d** Different constructs of human TNKS1 were expressed in HeLa cells. After TAP, the binding of each construct to endogenous SSSCA1 was assessed by western blotting. Green stars indicate the constructs of interest. **e** TNKS1 ARC1–3 was expressed and purified from *E. coli*. Schematic representation of the construct and Coomassie stained SDS-gel control after purification. **f** TNKS1 ARC1–3 binding to FAM-tagged peptides (positive control Axin1 in black, negative control Axin1 G>R in gray, SSSCA1-H1 in pink, and SSSCA1-H2 in purple) was assessed by fluorescence polarization. Affinities of each peptide for TNKS1 ARC1–3 are indicated. $n = 4$; error bars, SEM; $K_d$ error values, standard error of the fit; n.q. not quantifiable.

the core antiparallel β-sheet of the zinc-binding domain shows closest structural resemblance to the E3-ubiquitin-protein ligase Smad ubiquitination regulation factor 1 (SMURF1)[38] (PDB ID 2LAZ) with an overall root-mean-square deviation (RMSD) of 2.2 Å (Fig. 4d), the neuronal protein FE65[39] (PDB ID 2IDH) with

a RMSD value of 2.3 Å (Fig. 4e), and the ubiquitin ligase NEDD4 with a RMSD of 2.4 Å[40] (PDB ID 4N7F). The helical dimer interface has not been observed earlier for this type of Zn-binding proteins and the closest zinc-binding protein relative from the zinc ribbon domain 1 family (ZNRD1) do not share this. Only

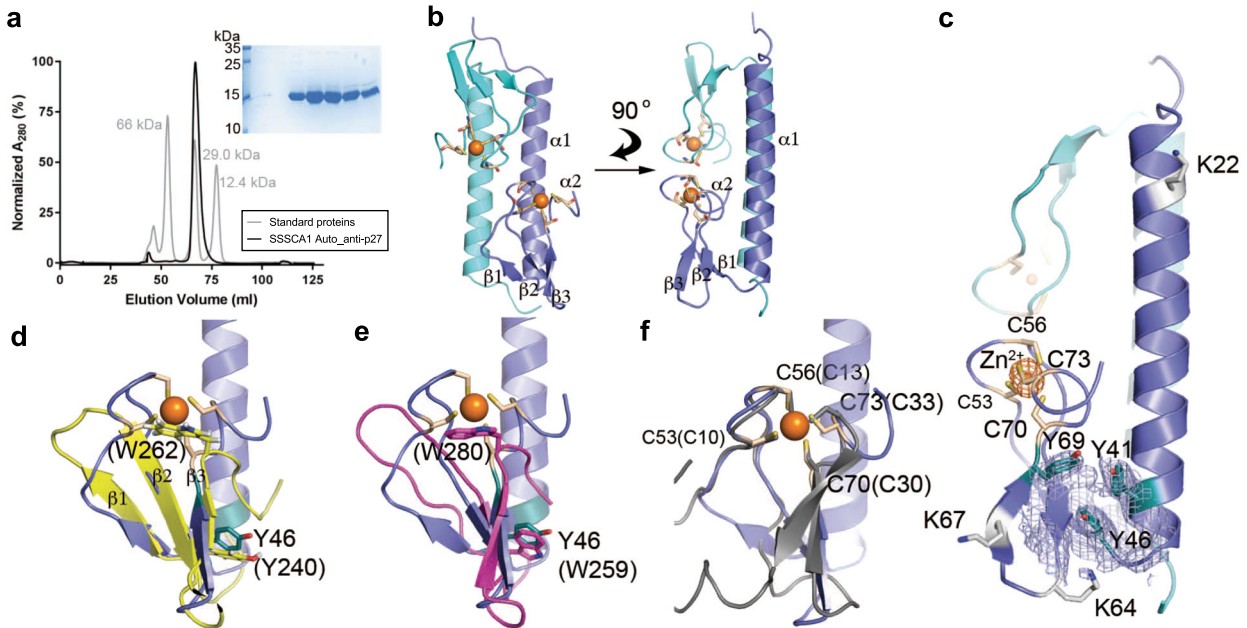

**Fig. 4 SSSCA1 belongs to a new family of Zn-binding proteins. a** SEC profiles of the N-terminal domain of SSSCA1 (black) and a set of three standard proteins (gray). The purified SSSCA1 N-terminal Auto_anti-p27 domain was subjected to SDS–PAGE and stained with Coomassie blue. **b** Crystal structure of the Auto_anti-p27 domain of human SSSCA1 at 2.3 Å resolution representing a new class of Zn-binding ribbon domain. Zinc ions are represented as orange spheres. Coordinating cysteine residues of each protomer are represented as sticks (wheat color). **c** Highly conserved cysteine residues are indicated as wheat colored sticks on the magnification of the Zn-binding site (orange sphere). The anomalous difference density map (orange map) is contoured at 5 σ and cover the Zn position. Lysine residues involved with ubiquitination are shown in gray and the conserved tyrosine cluster in dark cyan. An electron density map (2Fo-Fc, purple map) contoured at 1 σ is covering the tyrosine positions. The closest structure homologs of the core antiparallel β-sheet of the zinc-binding domain are two WW(Y) domains with conserved Trp and Tyr residues (sticks) highlighted: (**d**) The E3-ubiquitin-protein ligase SMURF1[38] (PDB ID 2LAZ) in yellow, and (**e**) the neuronal protein FE65[39] (PDB ID 2IDH) in magenta. **f** The C4 zinc-binding loops of the related Zn-binding ribbon domain 1 (ZNRD1) (gray) only have the position of the cysteines in common.

## Table 1 Data collection and refinement statistics.

|  | hSSSCA1 |
|---|---|
| Data collection | PETRA III |
| Space group | P 43 21 2 |
| Cell dimensions |  |
| $a$, $b$, $c$ (Å) | 49.49, 49.49, 164.18 |
| $\alpha$, $\beta$, $\gamma$ (°) | 90.0, 90.0, 90.0 |
| Wavelength (Å) | 1.2395 |
| Resolution (Å) | 47–2.3 (2.36–2.3)[a] |
| $R_{sym}$ or $R_{merge}$ | 0.04 (>100) |
| $R_{pim}$ | 0.02 (0.57) |
| $CC1/2$ | 1.0 (0.74) |
| $I/\sigma I$ | 26.2 (1.4) |
| Completeness (%) | 99.3 (98.5) |
| Redundancy | 9.2 (6.9) |
| Refinement |  |
| Resolution (Å) | 25.0–2.3 |
| No. reflections | 89,502 |
| No. Unique reflections | 9709 |
| $R_{work}/R_{free}$ | 0.22/0.26 |
| No. atoms | 1081 |
| Protein | 1066 |
| Ligand/ion | 8 |
| Water | 7 |
| B-factors | 85.6 |
| Protein | 85.6 |
| Ligand/ion | 105.4 |
| Water | 70.3 |
| R.m.s. deviations |  |
| Bond lengths (Å) | 0.006 |
| Bond angles (°) | 1.06 |
| Ramachandran statistics |  |
| Favored (%) | 91.3 |
| Allowed (%) | 7.1 |
| Outliers (%) | 1.6 |

Data collected from a single crystal.
[a]Values in parentheses are for highest-resolution shell.

superimposition of the zinc-binding residues is possible, shown for the only known member of ZNRD1 found in chain I (PDB ID 4C2M) of the RNA polymerase I[41] (Fig. 4f).

**SSSCA1 C-terminal domain behaves as a nuclear export signal.** The helix H2 in SSSCA1 C-terminal domain is predicted to be an amphipathic helix that contains motifs necessary for nuclear export. Leucine and isoleucine residues are numerous and conserved in this region across SSSCA1 orthologs (Fig. 5a). In particular, two leucine-rich NESs are predicted above the confidence level in fission yeast SSSCA1 helical domain (RIKEN *Schizosaccharomyces pombe* Postgenome Database). Zebrafish and worm SSSCA1 also show NES predictions with scores >0.5 in H2 (Fig. 5b, NetNES server[42]). In human SSSCA1, residues 173–194 display a positive Leu-rich NES prediction although below the 0.5 threshold (Fig. 5b). Similar NES motifs were first identified in PKIα and HIV-1 Rev proteins[43]. They consist of several closely spaced leucines or other large hydrophobic residues, and mutations of critical leucines abolish the export activity[43,44]. We first verified nuclear export activity by confocal microscopy in HeLa cells and show that both endogenous and overexpressed human SSSCA1 display a cytoplasmic subcellular localization (Fig. 5c). We then generated several mutants of SSSCA1 by site-directed mutagenesis and confirmed that their localization was dependent on the presence of an intact H2 region. The deletion of H2 (SSSCA1-ΔH2, -Δ(H1–H2), -Δ(Pro-H2)) or the simultaneous mutation of all leucine residues to alanine (SSSCA1-H2_{Leu/Ala}) lead to the ubiquitous localization of SSSCA1 constructs (Fig. 5c, Supplementary Fig. 1). Conversely, the constructs lacking one or several of the other domains (SSSCA1-ΔZn, -Δ(Zn-Pro)) still strictly localized in the cytoplasm.

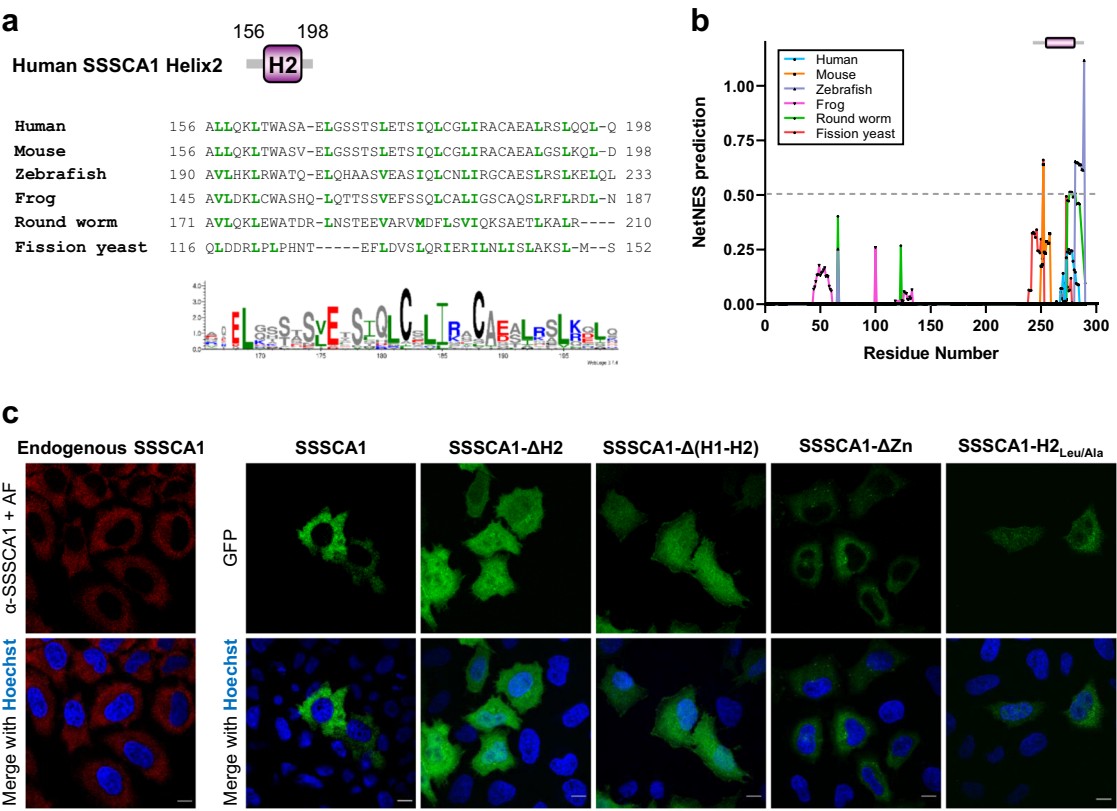

**Fig. 5 SSSCA1 C-terminal domain behaves as a nuclear export signal. a** Alignment of human (*Homo sapiens*), mouse (*Mus musculus*), zebrafish (*Danio rerio*), frog (*Xenopus laevis*), round worm (*Caenorhabditis elegans*) and fission yeast (*Sch. pombe*) SSSCA1 C-termini. Leucine and isoleucine residues are highlighted in green. WebLogo of SSSCA1 orthologs C-termini (human residues 156-198). **b** NetNES prediction plot of SSSCA1 orthologs. The gray bar indicates the residues of the alignment in **a**. **c** Confocal imaging of endogenous SSSCA1 and different GFP-tagged constructs of human SSSCA1 transiently expressed in HeLa cells. Scale bar: 10 μm.

## Discussion

The human *SSSCA1* gene is located on chromosome 11, one of the most gene- and disease-rich chromosomes in the genome[45]. In particular, the locus 11q13 where *SSSCA1* is located has been shown to be rearranged and amplified in multiple human cancers by numerous genetic mechanisms including deletions, translocations, loss of heterozygosity and allelic amplifications[46–50]. This could be one of the reasons why *SSSCA1* has been shown to be overexpressed in colorectal adenocarcinomas[10,11], identified as a target gene in breast cancer[12–14], and associated with disease progression and genomic instability in both metastatic oral squamous cell carcinoma[15] and pediatric metastatic neuroblastoma[16]. It is tempting to speculate that the reason could be that the oncogene c-MYC, altered in a variety of cancers, also regulates *SSSCA1*. We did observe that the knockdown of SSSCA1 slightly reduced the cell proliferation of the cells (Supplementary Fig. 2a, full blots in Supplementary Fig. 5e) and that c-MYC efficiently bound to the promoter region of the *SSSCA1* gene and regulated the expression at the mRNA and protein levels in the three cancer cell lines that were tested in this study (Supplementary Fig. 2b–d, full blots in Supplementary Fig. 5f).

The gene product of *SSSCA1* is a small soluble protein expressed in the cytoplasm of most cell types according to the Human Protein Atlas server[3]. We suggest that it is composed of three distinct domains and that this domain architecture is highly conserved throughout the eukaryote kingdom. Surprisingly, we could not identify any SSSCA1 orthologs in the bird and insect classes, where both the chicken and fruit fly genomes have been thoroughly annotated[51,52].

We initially set out experiments to confirm the putative binding of SSSCA1 with the human Tankyrases (TNKSs). Earlier proteomic data showed that SSSCA1 is pulled out in association with both TNKS1 and TNKS2[5–7]. It was however not certain whether SSSCA1 was a direct binding partner or simply part of a larger complex being pulled down with TNKS. We chose three different cancer cell lines (HeLa, SW480, and LNCaP) to increase the confidence level. However, compared with conventional immune precipitation, we made use of TAP to keep the level of false positives to a minimum and select for only the strongest binding partners (Fig. 4a). Using this method, we were able to use SSSCA1 to pull out specific binding partners. Unfortunately, the antibodies against MRM2 and VAPA gave unspecific results. The immunoblotting against γ-tubulin (TUBG1) was not conclusive (TUBG1 seemed to be pulled down by all SSSCA1 constructs and slightly by the GFP-tag alone) and the immunofluorescence showed that SSSCA1 and γ-tubulin did not co-localize at the centrosome (Supplementary Fig. 3a–b, full blots in Supplementary Fig. 5b, g). However, we cannot rule out that SSSCA1 and γ-tubulin are binding partners, for example during interphase. With this approach, we have therefore identified TNKS1 as a strong and confident binding partner of SSSCA1 in mammalian cells.

The enzyme TNKS1 and the closely related paralog TNKS2, which is less abundant, but was still identified in HeLa cells, are poly-ADP-ribosyltransferases involved in various processes such as telomere length regulation[53], mitotic spindle formation[54], proteasomal degradation[55,56], vesicle trafficking[57], and Wnt signaling pathway[6]. TNKS1 acts as a positive regulator of the Wnt signaling pathway by poly-ADP-ribosylation (PARylation) of

AXIN1 and AXIN2, two key components of the β-catenin destruction complex[6]. The inhibition of TNKS1 catalytic activity antagonizes Wnt signaling and has promising implications for cancer therapy, particularly colorectal cancer and non-small cell lung cancer[6,58,59]. Tankyrase is also crucial for embryonic development and adult homeostasis[60]. Understanding and characterizing the mechanisms that regulate TNKS activity are therefore of particular interest.

TNKS1 and TNKS2 share three conserved domains: the ARD responsible for substrate recruitment, an SAM domain that mediates homo- and hetero-oligomerization, and a C-terminal catalytic PARP domain[32,61]. The ARD is segmented into five conserved subdomains, or ARCs, which serve as a docking site for a wide range of target proteins and their subsequent scaffolding and/or PARylation. So far, most identified TNKS1 binding partners recognize various ARCs with a degenerate peptide motif, so-called Tankyrase-binding motif (TBM), with the consensus sequence R-x-x-[G or small hydrophobic]-[D/E]-G-[no P]-[D/E][61]. Here we have shown that the C-terminal helix H2 (residues 173–198) of SSSCA1 is critical for the binding of Tankyrase 1 ARC2 domain. Interestingly, the primary structure of SSSCA1 does not show any conserved TBM. The helix H2 in SSSCA1 contains a glycine (G182) and two arginine residues (R185 and R192) but these residues are not conserved among SSSCA1 orthologs and mutating these arginines to alanine residues do not alter the binding to TNKS1 (Supplementary Fig. 4, full blots in Supplementary Fig. 5c, h). This, to the best of our knowledge, could suggest that a novel binding motif between SSSCA1 and TNKS1 has been identified and that this binding mechanism needs to be investigated in the future. The protein binding between SSSCA1 and TNKS1 is likely an interaction specific to higher eukaryotes, since there are no TNKS homologs in the lower eukaryotes, like fission yeast. The interaction partner of fission yeast and other lower eukaryote SSSCA1 homologs will therefore remain unknown, but could still be an ARC like domain-containing protein, of which there are numerous.

The N-terminal zinc-binding domain shows a very high degree of conservation. We have determined the structure of this Cys4 zinc-binding domain and discovered that it belongs to a new family of undescribed zinc-binding proteins denominated Auto_anti-p27 (Pfam PF06677). This type of domain is predominantly found as a single motif in SSSCA1 orthologs (77%) in eukaryotes but also as uncharacterized proteins in archaea (22%) and bacteria (1%) (Pfam database[25]). It is exclusively associated with low complexity and disordered regions and its function is to date unknown. Our systematic search for protein with a similar structure to Auto_anti-p27 identified the HECT-type E3 SMURF1[38], the neuronal protein FE65 implicated in brain development and amyloid precursor protein signaling[39] and the NEDD4 ubiquitin-protein ligase recruited by arrestin-related domain-containing protein-3[40]. All three closest matches contain WW domains[62] which are characterized by their two highly conserved tryptophan residues that mediate binding with particular proline-rich peptides and/or phosphoserine- phosphothreonine-containing motifs. This would suggest that Auto_anti-p27 is as well implicated in protein–protein interactions and perhaps even have an E3-ubiquitin ligase activity, which would unlock an additional level of TNKS regulation, however, this remains to be investigated in the future. The zinc-binding domain is also the site of the majority of the potential ubiquitination sites (K22, K64, K67, and K82) identified in human cells. An ubiquitination could displace the dimer, or otherwise modify the function of this domain and alter its interaction with different potential partners (Fig. 4c). Our efforts to show E3 ubiquitination activity connected with SSSCA1 was however not convincing and will require further experiments.

The less conserved intermediate proline-rich disordered domain is highly flexible and connects two protein binding

domains. The IDRs are of particular interest because of their overrepresentation in regulatory proteins, signaling pathways, and diseases[63–65]. Many of these domains and regions undergo a folding coupled to binding with one or several specific partners, suggesting an important functional role[66]. The hallmark of this type of interaction is often characterized by high specificity, however, with low affinity. In this way, the intrinsically disordered domain of SSSCA1 may be involved in the accommodation and regulation of several inter- and/or intra-molecular interactions. This particular structural property may be desirable in a dynamic system that relies on responsive interaction networks where reversible binding is crucial.

The C-terminal domain is composed of two conserved α-helices. We have shown that the last helix H2 is crucial for the binding of protein targets. This amphipathic helix also behaves as an NES and directs SSSCA1 localization in the cytosol of human cells. NES sequences are essential regulators of the nucleocytoplasmic shuttling and the subcellular location of proteins in relation to cancer, cell-cycle progression and cell differentiation[67,68]. Human SSSCA1-H2 does not fit the "classical" NES pattern $\Phi$-$X_{2,3}$-$\Phi$-$X_{2,3}$-$\Phi$-$X$-$\Phi$, where $\Phi$ denotes hydrophobic residues and $X$ any amino acid. However, newly discovered NES motifs are more diverse and often deviate from the established consensus sequence[69]. Once more, the specificity of SSSCA1 seems to be particularly relevant for further investigation and better understanding of the breath of NES sequences.

In this study, despite several attempts, SSSCA1 was never found to be PARylated by TNKS1 and could not be directly linked to altering the Wnt signaling pathway under the tested conditions. However, we revealed that SSSCA1 binds proteins linked to microtubules and mitosis and possesses a strong NES. Further work exploring whether SSSCA1 could be acting as a molecular chaperone implicated in the binding of TNKS1 and its re-localization to the cytoplasm after mitosis is ongoing.

## Methods

**Phylogenetic tree and disorder prediction plots**. SSSCA1 orthologs' sequences (*Homo sapiens* ENSG00000173465, *Macaca mulatta* ENSMMUG00000017162, *Bos Taurus* ENSBTAG00000003713, *Canis lupus familiaris* ENSCAFG00000013480, *Rattus norvegicus* ENSRNOG00000012736, *Mus musculus* ENSMUSG000000794 78, *Monodelphis domestica* ENSMODG00000009665, *Chrysemys picta bellii* XP_005308009.1, *Strongylocentrotus purpuratus* XP_792794.3, *Danio rerio* ENSDARG00000069109, *Salmo salar* NP_001134714.1, *Takifugu rubripes* ENSTRUG00000008927, *Xenopus tropicalis* ENSXETG00000008424, *Xenopus laevis* NP_001079517.1, *Petromyzon marinus* ENSPMAG00000009726, *Rhipicephalus pulchellus* L7M6Z5, *Crassostrea gigas* EKC32325, *Nematostella vectensis* XP_001633291, *Ciona savignyi* ENSCSAVG00000008672, *Amphimedon queenslandica* I1FLC9, *Caenorhabditis elegans* Y39A1A.3, *Dictyostelium discoideum* XP_646324.1, *S. pombe* NP_596406.1) were retrieved from Ensembl, Genbank or UniProt. Sequences were visualized in Jalview and aligned using T-coffee. A phylogenetic tree was then constructed using the neighbor-joining tree method and edited in Dendroscope. Prediction of protein disorder was done using PONDR-FIT[70]. The predictor is the newest development of the PONDR (Predictors Of Natural Disordered Regions) series and shows good accuracy.

**Cell culture**. HeLa "Kyoto" cells were grown in DMEM (Gibco) and LNCaP cells in RPMI, both supplemented with 10% fetal calf serum (FCS) and maintained at 37 °C, 5% $CO_2$. SW480 cells were grown in L-15 medium (ATCC) supplemented with 10% FCS and maintained at 37 °C. For viability assays, the amount of viable cells was determined using TACS MTT Cell Proliferation Assays (Trevigen, 4890–025-K) following the manufacturer's recommendations.

**Plasmids, constructs and transfection**. Human *SSSCA1* (SSSCA1 FL, residues A2-H199) was synthesized by GenScript and cloned, using the *XhoI* and *SalI* restriction sites, into pNGFP-EU mammalian expression vector[71] with an N-terminal 8x His and EGFP tag. The constructs SSSCA1-Δ(Zn-Pro) (residues 149–199) and SSSCA1-H2$_{Leu/Ala}$ (L180A, L183A, I184A, L191A, L194A, L197A) were generated by GenScript from the original construct SSSCA1 FL/pNGFP-EU. The constructs SSSCA1-Δ(Pro-H2) (residues 2–84), SSSCA1-Δ(midPro-H2) (residues 2–111), SSSCA1-Δ(H1–H2) (residues 2–147) and SSSCA1-ΔH2 (residues 2–172) were all generated by insertion of a stop codon in the construct SSSCA1 FL/pNGFP-EU using the QuickChange Lighting

Enzyme Site-Directed Mutagenesis kit (Agilent) and the following primer pairs (Sigma-Aldrich): SSSCA1-Δ(Pro-H2) sens 5′-gacgtggataaagataat**taa**gtctgaatgccc ag-3′ and antisens 5′-ctgggcattcagagc**tta**attatctttatccagtc-3′, SSSCA1-Δ(midPro-H2) sens 5′-ctgggctct**tgatagg**cgccccagc-3′and antisens 5′-gctggggcgc**catca**agagccag-3′, SSSCA1-Δ(H1–H2) sens 5′-gtgcctccaaataca**taa**gtcatggcctgcacac-3′ and antisens 5′-gtgt gcaggccatgac**tta**tgtatttggaggcac-3′, and SSSCA1-ΔH2 sens 5′-gctccagcacc**taa**ctgga-gactagcatc-3′ and antisens 5′-gatgctagtctccag**tta**ggtgctggagc-3′. Finally, the construct SSSCA1-ΔZn (residues 85–199) was generated by mutagenesis-deletion from SSSCA1 FL/pNGFP-EU using the QuickChange Lighting Enzyme Site-Directed Mutagenesis kit and the following primer pair (Sigma-Aldrich): sens 5′-cgagccgagaatctttattttcagg gc**ccc**gctctgaatgcccagg-3′ and antisens 5′-cctgggcattcagagc**gggg**ccctgaaaataaagattctcggc tcg-3′.

The human TNKS1/pEGFP-C1 construct (TNKS1 FL, residues 1–1327) was kindly provided by Prof. Harald Stenmark (The Norwegian Radium Hospital, Oslo, Norway). TNKS1 FL was cloned by GenScript, using the *XhoI* and *SalI* restriction sites, into pNGFP-EU mammalian expression vector with an N-terminal 8x His and EGFP tag. The constructs TNKS1-Δ(SAM-PARP) (residues 1–975), TNKS1-Δ(ARC1-ARC4) (residues 835–1327), TNKS1-Δ(ARC1-ARC5) (residues 1014–1327), TNKS1-Δ(ARC2-PARP) (residues 1–331), TNKS1-Δ(ARC3-PARP) (residues 1–502), and TNKS1-Δ(ARC4-PARP) (residues 1–634) were all generated by GenScript from TNKS1 FL/pNGFP-EU.

The plasmids were transiently transfected into the different cell lines with XtremeGene 9 transfection reagent (Roche), following the manufacturer's recommendation. The plasmids coding for the full length SSSCA1 were diluted 1:3 with an empty vector to avoid the overexpression and aggregation of the protein.

siRNA reverse transfections of the three cell lines were performed using Lipofectamine RNAiMAX Reagent (Invitrogen), according to the manufacturer's recommendation. All siRNA were purchased from Thermo Scientific Dharmacon: Non-targeting siRNA control (D-001810–10–05), human SSSCA1 siRNA (L-020119–00–0005), human c-MYC siRNA (L-003282–00–0005), human TNKS1 siRNA (L-004740–00–0005), and human TNKS2 siRNA (L-004741–00–0005).

**Tandem-affinity purification coupled to mass spectrometry (TAP–MS).** HeLa, LNCaP, or SW480 cells were seeded in four 15-cm dishes 24 h before transfection. Eighteen hours post transfection, the cells were washed twice with PBS and lysed for 30 min, 4 °C, in solubilization/lysis buffer (50 mM HEPES, pH 7.4, 150 mM NaCl, 5% glycerol, 0.5% NP40, 1 mM NaF, 1 mM Na₃VO₄, 0.5 mM TCEP, 1 µg/ml DNAse I, 50 mM imidazole, 1 mM PMSF and protease cocktail inhibitor (Roche)). To achieve a complete lysis, cells were sonicated for five cycles 30 s ON/30 s OFF in a BioRuptor (Diagenode). Insoluble material was removed by centrifugation for 15 min at 15,000 × g, 4 °C. The cleared lysates were loaded consecutively on His SpinTrap columns (GE Healthcare). The beads were washed three times with five bead volumes of equilibration buffer (50 mM HEPES, pH 7.4, 150 mM NaCl, 5% glycerol, 0.5% NP40, 50 mM imidazole) and the proteins complexes were eluted with five bead volumes of elution buffer (50 mM HEPES, pH 7.4, 150 mM NaCl, 5% glycerol, 0.5% NP40, 500 mM imidazole). The eluates were then incubated with 20 µl GFP-Trap A beads (ChromoTek) overnight at 4 °C on a rotation shaker. Immunoprecipitates were washed four times with 500 µl of equilibration buffer (without imidazole) and protein complexes were eluted by boiling 10 min at 95 °C in 100 µl of 2x Laemmli buffer. Samples were then loaded on a 12% SDS–PAGE gel and resolved by electrophoresis. Gel lanes were cut into slices and subjected to overnight in-gel tryptic digestion. Peptides were analyzed by an ESI-Orbitrap (LTQ Orbitrap XL, Thermo Scientific) mass spectrometer coupled to a nano-LC system. Peptides were purified by C18 ZipTips (Millipore) before injected into an Ultimate 3000 nanoLC system (Dionex). For separation of peptides an Acclaim PepMap 100 column (50 cm × 75 µm) packed with 100 Å C18 3 µm particles (Dionex) was used. A flow rate of 300 nL/min was employed with a solvent gradient of 7–35% B in 40 min, to 50% B in 3 min and then to 80% B in 2 min. Solvent A was 0.1% formic acid and solvent B was 0.1% formic acid/90% acetonitrile. The mass spectrometer was operated in data-dependent mode and survey full scan MS spectra were acquired in the Orbitrap with 60.000 resolution at m/z 400. The method allowed sequential isolation of the seven most intense ions in the linear ion trap using collision induced dissociation (CID). Target ions already selected for MS/MS were dynamically excluded for 60 s.

**Bioinformatics.** MS data were acquired using Xcalibur v2.5.5 and processed using Proteowizard v3.0.5047. Mascot v.2.3 was used as search engine and the data were searched against the human subset of SwissProt v.2011_11 using trypsin as enzyme with one missed cleavage allowed. Variable modifications allowed included protein N-terminal acetylation, methionine oxidation, pyro-glutamic acid for N-terminal glutamines, propionamide on cysteines, deamidation of glutamines and asparagines as well as phosphorylation of serines, threonines and tyrosines. Mascot results were imported into Scaffold v3.6 with the identification criteria: minimum two peptides per protein, peptide probability >90% and protein probability >99%. This yielded protein FDRs at 0.2%. As quantitative value, emPAI values[72] were calculated based on the number of unique peptides listed in Scaffold by an in-house script.

**Western blotting.** Proteins complexes obtained after TAP were boiled once more for 3 min at 95 °C and loaded on a 12% SDS–PAGE gel. Proteins were transferred to PVDF membranes and incubated for 1 h at room temperature in TBS-0.05% Tween 20 (TBS-T)-5% BSA or -5% nonfat dry milk. The membranes were probed overnight at 4 °C with the following primary antibodies: mouse anti-SSSCA1 (1:2000, 2F5, Novus Biologicals), rabbit anti-TNKS 1/2 (1:200, sc-8337, Santa Cruz Biotechnology), rabbit anti-GFP tag (1:1000, #2555, Cell Signaling Technology), mouse anti-β-actin (1:1000, #3700, Cell Signaling Technology), rabbit anti-c-MYC (1:1000, ab32072, Abcam), mouse anti-γ-tubulin (1:1000, NB100–74421, Novus Biologicals), diluted in TBS-T-5% BSA or −5% nonfat dry milk. Primary antibodies were visualized with secondary HRP-conjugated antibodies (Cell Signaling Technology) and Chemilumi-nescence Blotting substrate (Roche) on a Chemidoc Imaging System (Bio-Rad).

**Immunofluorescence microscopy.** HeLa cells were seeded in 24-well plates on glass slides 24 h before transfection. Eighteen hours post transfection, the cells were washed once in PBS before fixation for 20 min in PBS-3% paraformaldehyde. Fixed cells were washed with PBS and quenched for 10 min with PBS-50 mM NH₄Cl. Fixed cells were then permeabilized for 10 min with PBS-0.25% Saponin. Cells were incubated overnight at 4 °C in a wet chamber with a 1:100 dilution of a mouse anti-SSSCA1 antibody (2F5, Novus Biologicals) or a 1:200 dilution of rabbit anti-TNKS1/2 antibody (sc-8337, Santa Cruz Biotechnology) or a 1:50 dilution of mouse anti-γ-tubulin (NB100–74421, Novus Biologicals) in PBS-0.25% Saponin. Cells were washed three times with PBS-0.25% Saponin and stained for 20 min in a wet chamber with 1:200 dilution of anti-mouse-AlexaFluor 594 or anti-rabbit-AlexaFluor 680 secondary antibody (Invitrogen). After three washes with PBS-0.25% Saponin, the nuclei were stained for 8 min with 2 µg/ml Hoechst 33258 (Invitrogen). Finally, cells were washed with PBS, quickly rinsed in H₂O and mounted in mowiol 4–88. Images were obtained on LSM 510 (Zeiss) microscope with a 63 × 0.55 numerical aperture oil objective.

**Fluorescence polarization binding assays.** Human TNKS1's ARC subdomains 1–3 (ARC1–3, residues 178–645) were cloned from TNKS1 FL/pNGFP-EU into pETM11 bacterial expression vector by GenScript, using the *NcoI* and *XhoI* restriction sites. We purified TNKS1 ARC1–3 from *E. coli*, following the protocol of Eisemann et al. [73]. Binding reactions were performed in a total volume of 50 µL with 50 nM 5-FAM-Ahx-conjugated peptides and the indicated concentrations of ARC1–3 protein in FP buffer (25 mM HEPES pH 8, 150 mM NaCl, 1 mM TCEP, 5% glycerol, and 0.05% CHAPS). Mixtures were incubated at room temperature for 60 min in a Nunclon Delta Surface black 96-well plate (Thermo Fisher Scientific). FP measurements were read on a Jasco FP-8500 Spectrofluorometer with an excitation of 470 nm and an emission of 520 nm. Means of four replicates ($n = 4$) and standard error of the mean (SEM) are shown. Nonlinear regression with FP values was performed in GraphPad Prism 7.00 (GraphPad Software, La Jolla, California, USA) using a one-site total binding model. All conjugated peptides were purchased from GenScript: Axin1 positive control (EDAPRPPVPGEEG), Axin1 G>R negative control (EDAPRPPVPREEG), SSSCA1-H1 (DVMACTQTALLQKL TWASAELGSS), and SSSCA1-H2 (TSLETSIQLCGLIRACAEALRSLQQL).

**Cloning, expression and purification of SSSCA1 Auto_anti-p27.** The N-terminal-domain of human SSSCA1 (Auto_anti-p27 domain, M1-S111) was syn-thesized and codon optimized for *E. coli* expression by GenScript. Auto_anti-p27 was cloned into a pETM11 vector using the *NcoI* and *XhoI* restriction sites, with an N-terminal 6x His tag and a tobacco etch virus (TEV) protease cleavage site. The plasmid was transformed into *E. coli* Rosetta 2 cells (Novagen). Colonies were inoculated in standard Luria-Bertani (LB) medium overnight (o/n) at 37 °C, 200 rpm, containing 35 µg/mL chloramphenicol (Cam) and 50 µg/mL kanamycin (Kan). Fresh o/n cultures were used to inoculate 3 L LB medium supplemented with 35 µg/mL Cam, 50 µg/mL Kan and 500 µL/L polyprolyne glycol P2000 anti-foam (Sigma-Aldrich). The culture was grown at 37 °C, with full airflow in a Lex HT Bioreactor (Harbinger), until $OD_{600}$ reached 0.6–0.8. The culture was cool-downed on ice and expression of Auto_anti-p27 was induced by addition of 1 mM IPTG. The growth continued overnight at 18 °C in the Fermentor. Cells were harvested by centrifugation at 10,000 × g for 25 min, 4 °C. The resulting cell pellet was resuspended in washing buffer (20 mM HEPES pH 7.4, 500 mM NaCl, 10 % glycerol, 0.5 mM TCEP) supplemented with 1 mM freshly added PMSF and cen-trifuged at 4000 × g for 15 min, 4 °C. An eight times weight volume lysis buffer (20 mM HEPES pH 7.4, 1 M NaCl, 10% glycerol, 1 M urea, 0.1% NP40, and 0.5 mM TCEP) supplemented with 1 mM PMSF and 5 µg/mL DNase I was added to 19–22 g wet weight of cells. Cells were lysed using a High Pressure Homogenizer C3 (Avestin) by three runs at 15,000–20,000 p.s.i. Cell debris was removed by cen-trifugation at 40,000 × g for 30 min at 4 °C.

Protein purification was performed using ÄKTAprime Plus for the affinity purification (GE Healthcare) and ÄKTApurifier for the gel filtration (GE Healthcare). The cleared lysate was supplemented with 20 mM imidazole, filtered on a 0.45 µm membrane and loaded on a HisTrap FF 5 ml column (GE Healthcare), preequilibrated in calibration buffer (20 mM HEPES pH 7.4, 500 mM NaCl, 10% glycerol, 20 mM imidazole, and 0.5 mM TCEP). The Ni²⁺-column was then washed with calibration buffer (8 column volumes, CV). Bound proteins were eluted with an imidazole gradient from 20 to 500 mM in 1.5 mL fractions. Fractions were analyzed by

SDS–PAGE and target fractions were pooled. To cleave the His tag, GFP-TEV protease was added at 0.05 mg/mL and incubated for 16 h in the cold room on a magnetic stirrer. After dilution to obtain a final concentration of 20 mM imidazole, the noncleaved Auto_anti-p27 was removed by applying the protein solution to the Ni$^{2+}$-column again and collecting the flow-through.

The flow-through of the second IMAC was concentrated down to 1–2 mL at 10–15 mg/mL before being applied to a HiLoad Superdex S75 16/600 PG 120 ml gel filtration column (GE Healthcare) equilibrated in separation buffer (20 mM HEPES pH 7.4, 200 mM NaCl, 5% glycerol, and 0.1 mM TCEP). The protein was eluted with 1 CV of separation buffer in fractions of 1.5 mL and the purified protein was analyzed again by SDS–PAGE. Target fractions were pooled and concentrated above 30 mg/mL using a VivaSpin 6 10,000 MWCO centrifugal filter device (Sartorius Stedim Biotech). Protein concentration was measured by absorption at a wavelength of 280 nm using a calculated extinction coefficient on a NanoDrop 2000 Spectrophotometer (Thermo Fisher Scientific). The purified protein solution was aliquoted, flash freezed with liquid nitrogen and stored at −80 °C until further use.

To estimate the native molecular weight of Auto_anti-p27, the common standard proteins (MWGF200-1KT, Sigma-Aldrich) cytochrome C (12.4 kDa), carbonic anhydrase (29 kDa), and albumin serum bovine (66 kDa) were run on the HiLoad Superdex S75 16/600 PG column using the same buffer and conditions.

**Crystallization and data collection**. An aliquot of 100 µL of purified and concentrated SSSCA1 Auto_anti-p27 protein was thawed, buffer exchanged against a dialysis buffer (2 mM HEPES pH 7.4, 100 mM NaCl) using a Vivaspin 500–10,000 MWCO and concentrated between 50 and 95 mg/mL. Crystals were obtained by the sitting drop vapor diffusion method in a 24-well plate. A volume of 1 µL protein solution was mixed 1:1 with the reservoir solution (3.5 M sodium formate pH 7.0, 10% glycerol) and the plates were left to incubate at 18 °C. Crystals appeared after ~1 week and continued to grow for ≥2 weeks more. Crystals were quickly frozen in liquid nitrogen.

Crystals were mounted in LithoLoops (Molecular Dimensions) and cryo-cooled in liquid nitrogen. An initial complete dataset was collected at 100 K on beamline ID23 at the European Synchrotron Radiation Facility (Grenoble, France) at 2.8 Å resolution and optimized dataset data extended to 2.3 Å on the P14 beamline at PETRAIII in Hamburg, Germany. The data were processed with XDS. The structure was solved with single-wavelength anomalous diffraction by Phaser[74]. Model building and refinement were performed using Coot[75] and phenix.refine within the Phenix package[76], the refinement was finalized in Refmac[77]. NCS restraints were only used initially as it became evident that there were minor molecular differences between the two protomers, likely induced by crystal contacts. All structure validations were performed using MolProbity[78]. Structure analysis and figure preparation were done using PyMOL Molecular Graphics System, Version 1.5.0.3, Schrödinger, LLC. The finalized model and structure factors were deposited to PDB and given the PDB accession code (6HCZ).

**Statistics and reproducibility**. Unless otherwise noted, experiments were repeated at least three times. All data are presented as means ± SEM or standard deviation, as indicated for each figure. Statistical analyses between groups were determined by unpaired t test in GraphPad Prism 7.00 (GraphPad Software, La Jolla, California, USA). A p value of <0.05 was considered statistically significant.

**Reporting summary**. Further information on research design is available in the Nature Research Reporting Summary linked to this article.

## Data availability
Data that support the findings of this study have been deposited in protein data bank with the PDB identifier code 6HCZ. Additional data are implemented as Supplementary data. All other relevant data are available from the corresponding author.

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

## Acknowledgements

H.P.D. was supported by the Norwegian Cancer Society (grant no. 4483570-2013). S.K. was supported by the Research Council of Norway (grants no. 262613 and 174938/030). We thank Jo Waaler for giving us an aliquot of the SW480 cells and protocols on how to culture them.

## Author contributions

H.P.D. and J.P.M. conceived and planned the experiments, and analyzed the data. H.P.D. performed the experiments. J.P.M. carried out the crystal data collection and structure determination. C.P. contributed to the confocal microscopy. S.J.B. performed the siRNA and ChIP experiments. H.G. helped with the cell culture experiments. B.T. and M.A. performed the mass spectrometry study and the bioinformatic analysis. O.B., I.G.M., and S.K. provided critical feedback. H.P.D. and J.P.M. wrote the paper with input from all authors.

## Competing interests

The authors declare no competing interests.
