## [Peer Review File · Communications Biology]

Reviewers' comments:

Reviewer #1 (Remarks to the Author):

In this study, Perdreau-Dahl et al. systematically characterized the SSSCA1 protein domains, binding partners, cellular localization and potential roles in cancer cells. This study may provide a useful resource for the researchers in the related fields, although the functional relevance is still largely unknown.

Major points:

1. In Fig2, the authors only validated the interaction between SSSCA1 and Tankyrase1, which actually has been reported by others before. How about other identified binding proteins in the SSSCA1 TAP-MS analysis?
2. The authors showed that SSSCA1 can be tyrosine phosphorylated (Fig2e) and proposed Tyr69 as the phosphorylation site. This conclusion is required to be at least validated by using the SSSCA1 Y69F mutant. If SSSCA1 can be phosphorylated at Tyr69, will the phosphorylation affect the Zn-binding domain dimer formation?

Minor points:

1. The authors may consider including sufficient labeling in the figures to help readers interpret the data more easily. For example, Fig1c shows species in different colors but did not explain what the colors mean; it is hard to tell what the curves/bars represent in Fig3e, FigS1a, FigS1b and FigS1d without checking their legends.
2. Please provide loading input controls for both Fig3b and Fig3d.
3. Scale bar is required for all the IF figures.

Reviewer #2 (Remarks to the Author):

SSSCA1 is a potential player in Sjögren syndrome and a number of cancers. The authors conducted extensive characterization of SSSCA1 and provided some important clues about the potential functions of this protein. 1). The authors conducted thorough proteomic analysis of SSSCA1 and identified conserved domains of the protein. 2). They conducted GFP and HIS pull-down and MS studies and observed interactions between SSSCA1 and TNKS1. Next they conducted truncation, co-IP and fluorescence polarizations experiments and identified the domains of SSSCA1 and TNKS1 that mediate their interactions. 3). They expressed the N-terminal zinc finger domain of SSSCA1 and determined the crystal structure of that domain and observed a novel zinc finger like fold. 4). They identified that the C-terminal H2 domain of SSSCA1 contains a nuclear export signal that target SSSCA1 to the cytosol. Overall these studies are pretty solid with various kinds of data to support the findings. I think this work is significant and should be considered for publication with some minor revisions.

The major shortcoming of this work is that the biological function of SSSCA1 was not well established. For example, it is not clear how this protein regulates Wnt signaling. Although the structure of the SSSCA1 zinc finger domain is interesting, but it feels like an isolated piece of study without much functional connections. Since the zinc finger domain has structural similarities to E3 ligase SMURF1, the function of that domain can be explored to support the nice structural studies. It can be challenging to figure out the exact function of SSSCA1. Considering the amount of the solid data provided here, this reviewer supports the consideration of this work for publication and leave the functional studies to future work.

There are a number of minor problems in spelling, such 1990's (line 28, 44 introduction), DNase, xg, etc (line 528, 532 in methods) that need to be revised. The method for structural determination should be "single wavelength anomalous diffraction" instead of "single anomalous diffraction" (line 575). "Crystallography" in Line 515 seems need to be removed from title of that method section. I encourage the authors to polish the manuscript carefully to make sure it read smoothly.

Reviewers' comments:

Reviewer #1 (Remarks to the Author):

In this study, Perdreau-Dahl et al. systematically characterized the SSSCA1 protein domains, binding partners, cellular localization and potential roles in cancer cells. This study may provide a useful resource for the researchers in the related fields, although the functional relevance is still largely unknown.

Major points:

1. In Fig2, the authors only validated the interaction between SSSCA1 and Tankyrase1, which actually has been reported by others before. How about other identified binding proteins in the SSSCA1 TAP-MS analysis?

>>We have included our validation data for TUBG1, which we thought could be a potential binding partner, we have taken a conservative approach in this study, which does not exclude other binding partners.

2. The authors showed that SSSCA1 can be tyrosine phosphorylated (Fig2e) and proposed Tyr69 as the phosphorylation site. This conclusion is required to be at least validated by using the SSSCA1 Y69F mutant. If SSSCA1 can be phosphorylated at Tyr69, will the phosphorylation affect the Zn-binding domain dimer formation?

>>We have removed the story about the phosphorylated tyrosine as it did not bring that much information as it was presented in the present manuscript. We actually agree with the reviewer that to keep it in would require more validation. We therefore decided to remove it.

Minor points:

1. The authors may consider including sufficient labelling in the figures to help readers interpret the data more easily. For example, Fig1c shows species in different colours but did not explain what the colours mean; it is hard to tell what the curves/bars represent in Fig3e, FigS1a, FigS1b and FigS1d without checking their legends.

>> In Fig3f we have added an explanation and the Apparent Kd values.
We have added an additional FigS1 with additional confocal pictures and GFP-tag control.
We have added explanations to Fig4a, FigS2a, FigS2b and FigS2d.

2. Please provide loading input controls for both Fig3b and Fig3d.

>>fixed

3. Scale bar is required for all the IF figures.

>>fixed

Reviewer #2 (Remarks to the Author):

SSSCA1 is a potential player in Sjögren syndrome and a number of cancers. The authors conducted extensive characterization of SSSCA1 and provided some important clues about the potential functions of this protein. 1). The authors conducted thorough proteomic analysis of SSSCA1 and identified conserved domains of the protein. 2). They conducted GFP and HIS pull-down and MS studies and observed interactions between SSSCA1 and TNKS1. Next they conducted truncation, co-IP and fluorescence polarizations experiments and identified the domains of SSSCA1 and TNKS1 that mediate their interactions. 3). They expressed the N-terminal zinc finger domain of SSSCA1 and determined the crystal structure of that domain and observed a novel zinc finger like fold. 4). They identified that the C-terminal H2 domain of SSSCA1 contains a nuclear export signal that target SSSCA1 to the cytosol. Overall these studies are pretty solid with various kinds of data to support the findings. I think this work

is significant and should be considered for publication with some minor revisions.

The major shortcoming of this work is that the biological function of SSSCA1 was not well established. For example, it is not clear how this protein regulates Wnt signaling. Although the structure of the SSSCA1 zinc finger domain is interesting, but it feels like an isolated piece of study without much functional connections. Since the zinc finger domain has structural similarities to E3 ligase SMURF1, the function of that domain can be explored to support the nice structural studies. It can be challenging to figure out the exact function of SSSCA1. Considering the amount of the solid data provided here, this reviewer supports the consideration of this work for publication and leave the functional studies to future work.

>>We appreciate the comments made and are grateful that our hard work has been recognized, we did spend several years trying to find the function, that indeed would have been a satisfying conclusion of the project. We at this point decided not to sit on the data anymore, as this paper focusses on the interaction with Tankyrase, major function of SSSCA1 is likely more diverse.

There are a number of minor problems in spelling, such 1990's (line 28, 44 introduction), >>fixed

DNase, xg, etc (line 528, 532 in methods) that need to be revised.

>>fixed

The method for structural determination should be "single wavelength anomalous diffraction" instead of "single anomalous diffraction" (line 575).

>>fixed

"Crystallography" in Line 515 seems need to be removed from title of that method section. I encourage the authors to polish the manuscript carefully to make sure it read smoothly.

>>We have careful read through the manuscript and believe it reads more smoothly now.

REVIEWERS' COMMENTS:

Reviewer #1 (Remarks to the Author):

The authors have fully addressed my previous questions about the manuscript. Thanks!